# THE POLARISED REGIME OF IDENTIFIABLE VARIATIONAL AUTOENCODERS

**Lisa Bonheme & Marek Grzes**
School of Computing
University of Kent
Canterbury, UK
{lb732, m.grzes}@kent.ac.uk

## ABSTRACT

The polarised regime—the capacity of variational autoencoders (VAEs) to discard superfluous latent variables—is well-studied in the context of "classical" VAEs with a standard Gaussian prior. In this paper, we extend these results to the case of identifiable VAEs (iVAEs).

**Motivation for this study**  The polarised regime of VAEs gives them a signature behaviour with specific strengths and weaknesses which are well-studied (Dai et al., 2017; Rolinek et al., 2019; Dai et al., 2020; Bonheme & Grzes, 2021). In this paper, we show that iVAEs also behave in a polarised regime and are thus likely to exhibit the same behaviour.

## 1 BACKGROUND

**Variational Autoencoders**  VAEs (Kingma & Welling, 2014) are non-identifiable deep probabilistic generative models which maximise $\mathcal{L}(\boldsymbol{\theta}, \boldsymbol{\phi}; \mathbf{x}) = \mathbb{E}_{q_{\boldsymbol{\phi}}(\mathbf{z}|\mathbf{x})}[\log p_{\boldsymbol{\theta}}(\mathbf{x}|\mathbf{z})] - D_{\mathrm{KL}}(q_{\boldsymbol{\phi}}(\mathbf{z}|\mathbf{x}) \parallel p(\mathbf{z}))$ where $p(\mathbf{z}) = \mathcal{N}(\mathbf{0}, \boldsymbol{I})$.

**Identifiable Variational Autoencoders**  iVAEs (Khemakhem et al., 2020) are identifiable versions of VAEs with a conditionally factorial prior $p_{\boldsymbol{\theta}}(\mathbf{z}|\mathbf{u})$ where $\mathbf{u}$ is observed. They maximise $\mathcal{L}(\boldsymbol{\theta}, \boldsymbol{\phi}; \mathbf{x}, \mathbf{u}) = \mathbb{E}_{q_{\boldsymbol{\phi}}(\mathbf{z}|\mathbf{x}, \mathbf{u})}[\log p_{\boldsymbol{\theta}}(\mathbf{x}|\mathbf{z})] - D_{\mathrm{KL}}(q_{\boldsymbol{\phi}}(\mathbf{z}|\mathbf{x}, \mathbf{u}) \parallel p_{\boldsymbol{\theta}}(\mathbf{z}|\mathbf{u}))$. Their natively disentangled latent representations are semantically meaningful, fair, and beneficial for abstract reasoning (Locatello et al., 2019; van Steenkiste et al., 2019), which makes them attractive for downstream tasks.

**The polarised regime**  The polarised regime is the capacity of any well-behaved VAE to discard superfluous (passive) latent variables while learning the remaining active variables with high precision (Rolinek et al., 2019; Dai & Wipf, 2018; Dai et al., 2018). When this is extended to multiple inputs, variables can be mixed, active, or passive (Bonheme & Grzes, 2021). Mixed if they are active for some inputs and passive for others, active (or passive) if they are active (or passive) for all inputs.

**Notation**  Given $\boldsymbol{\epsilon}$ sampled from $\mathcal{N}(\mathbf{0}, \boldsymbol{I})$, the mean, variance and sampled representations of the encoder are defined as $\boldsymbol{\mu} \triangleq \boldsymbol{\mu}(\mathbf{x}, \mathbf{u}; \boldsymbol{\phi})$, $\boldsymbol{\sigma} \triangleq diag[\boldsymbol{\Sigma}(\mathbf{x}, \mathbf{u}; \boldsymbol{\phi})]$, and $\mathbf{z} \triangleq \boldsymbol{\mu} + \boldsymbol{\epsilon}\boldsymbol{\sigma}^{1/2}$, such that $q_{\boldsymbol{\phi}}(\mathbf{z}|\mathbf{x}, \mathbf{u}) = \mathcal{N}(\boldsymbol{\mu}, diag[\boldsymbol{\sigma}])$. The prior representations are denoted by $^{\dagger}$, such that $p_{\boldsymbol{\theta}}(\mathbf{z}|\mathbf{u}) = \mathcal{N}(\boldsymbol{\mu}^{\dagger}, diag[\boldsymbol{\sigma}^{\dagger}])$. Specific samples of the observations $\boldsymbol{X} \triangleq \{\mathbf{x}^{(i)}\}_{i=1}^{n}$ and $\boldsymbol{U} \triangleq \{\mathbf{u}^{(i)}\}_{i=1}^{n}$ will be indicated by $^{(i)}$ such that $\boldsymbol{\mu}^{(i)} \triangleq \boldsymbol{\mu}(\mathbf{x}^{(i)}, \mathbf{u}^{(i)}; \boldsymbol{\phi})$, and the $j^{th}$ dimension of a vector by $_j$ (e.g., $\boldsymbol{\mu}_j$).

## 2 EXTENSION OF THE POLARISED REGIME TO IVAES

**Assumptions**  We consider iVAEs with a Gaussian prior and posterior, similarly to Khemakhem et al. (2020), where the mean and variance of the prior are learned (e.g., by a deep neural network). We also assume that the number of samples $n$ is sufficiently large for sample mean and variance to be good approximations of the true mean and variance.

The proofs and an empirical verification of this section can be found in Appendices A and B.

## 2.1 iVAEs learn in a polarised regime

Extending the work of Dai et al. (2018) to iVAEs, we have:

**Theorem 1** (Polarised regime of iVAEs). *Any well-behaved iVAE learns in a polarised regime.*

Given Theorem 1, one can then readily extend the definition of Rolinek et al. (2019) to iVAEs.

**Proposition 1** (Polarised regime of $\boldsymbol{\mu}^{(i)}$, $\boldsymbol{\sigma}^{(i)}$, and $\mathbf{z}^{(i)}$). *When an iVAE learns in a polarised regime, its mean, variance, and sampled representations, $\boldsymbol{\mu}$, $\boldsymbol{\sigma}$, and $\mathbf{z}$, are composed of a set of passive and active variables, $\mathbb{V}_p^{(i)} \cup \mathbb{V}_a^{(i)}$ such that, for each pair of $\mathbf{x}^{(i)}$ and $\mathbf{u}^{(i)}$:*

*(i)* $\boldsymbol{\mu}_j^{(i)} \approx \boldsymbol{\mu}_j^{\dagger(i)}$, $\boldsymbol{\sigma}_j^{(i)} \approx \boldsymbol{\sigma}_j^{\dagger(i)}$, and $\mathbf{z}_j^{(i)} \approx \mathbf{z}_j^{\dagger(i)}$ $\quad \forall\, j \in \mathbb{V}_p^{(i)}$,

*(ii)* $\boldsymbol{\sigma}_j^{(i)} \ll 1$ and $\mathbf{z}_j^{(i)} \approx \boldsymbol{\mu}_j^{(i)}$ $\quad \forall\, j \in \mathbb{V}_a^{(i)}$,

*where $\boldsymbol{\mu}^\dagger \triangleq \boldsymbol{\mu}^\dagger(\mathbf{u}; \boldsymbol{\theta})$, $\boldsymbol{\sigma}^\dagger \triangleq diag[\boldsymbol{\Sigma}^\dagger(\mathbf{u}; \boldsymbol{\theta})]$, and $\mathbf{z}^{\dagger(i)} = \boldsymbol{\mu}^{\dagger(i)} + \boldsymbol{\epsilon}^{(i)} \left(\boldsymbol{\sigma}^{\dagger(i)}\right)^{1/2}$.*

## 2.2 Polarised regime of iVAEs for multiple data examples

Based on Proposition 1 and Bonheme & Grzes (2021), we will now consider the properties of the latent representations of iVAEs over multiple data examples. As for VAEs, the passive variables of the mean representation are always close to the mean of their corresponding prior.

**Proposition 2** (Polarised regime of $\boldsymbol{\mu}$ over $\boldsymbol{X}$ and $\boldsymbol{U}$). *When an iVAE learns in a polarised regime, its mean representation $\boldsymbol{\mu} \approx \boldsymbol{\mu}(\boldsymbol{X}, \boldsymbol{U}; \boldsymbol{\phi})$ is composed of a set of passive, active and mixed variables $\mathbb{V}_p \cup \mathbb{V}_a \cup \mathbb{V}_m$ such that, over $\boldsymbol{X}$ and $\boldsymbol{U}$:*

*(i)* $\bar{\boldsymbol{\mu}}_j \approx \bar{\boldsymbol{\mu}}_j^\dagger$ and $Var(\boldsymbol{\mu}_j) \approx Var(\boldsymbol{\mu}_j^\dagger)$ $\quad \forall\, j \in \mathbb{V}_p$,

*(ii) If the mean of the prior is fixed to some vector $\boldsymbol{\mu}^\dagger$ and only its variance is learned, then* $\bar{\boldsymbol{\mu}}_j \approx \boldsymbol{\mu}_j^\dagger$ and $Var(\boldsymbol{\mu}_j) \ll 1$ $\quad \forall\, j \in \mathbb{V}_p$,

*where $\boldsymbol{\mu}^\dagger \approx \boldsymbol{\mu}^\dagger(\boldsymbol{U}; \boldsymbol{\theta})$, $\bar{\boldsymbol{\mu}}_j \triangleq \mathbb{E}_{p(\mathbf{x},\mathbf{u})}[\boldsymbol{\mu}_j]$, $\bar{\boldsymbol{\mu}}_j^\dagger \triangleq \mathbb{E}_{p(\mathbf{u})}[\boldsymbol{\mu}_j^\dagger]$, and $Var(\cdot)$ denotes the variance.*

Moreover, the variance representation will be close to zero for active variables and to the prior's variance for passive variables to respectively maintain high precision and low KL divergence.

**Proposition 3** (Polarised regime of $\boldsymbol{\sigma}$ over $\boldsymbol{X}$ and $\boldsymbol{U}$). *When an iVAE learns in a polarised regime, its variance representation $\boldsymbol{\sigma} \approx diag[\boldsymbol{\Sigma}(\boldsymbol{X}, \boldsymbol{U}; \boldsymbol{\phi})]$ is composed of a set of passive, active and mixed variables $\mathbb{V}_p \cup \mathbb{V}_a \cup \mathbb{V}_m$ such that, over $\boldsymbol{X}$ and $\boldsymbol{U}$:*

*(i)* $\bar{\boldsymbol{\sigma}}_j \approx \bar{\boldsymbol{\sigma}}_j^\dagger$ and $Var(\boldsymbol{\sigma}_j) \approx Var(\boldsymbol{\sigma}_j^\dagger)$ $\quad \forall j \in \mathbb{V}_p$,

*(ii)* $\bar{\boldsymbol{\sigma}}_j \ll 1$ and $Var(\boldsymbol{\sigma}_j) \ll 1$ $\quad \forall j \in \mathbb{V}_a$,

*where $\boldsymbol{\sigma}^\dagger \approx diag[\boldsymbol{\Sigma}^\dagger(\boldsymbol{U}; \boldsymbol{\theta})]$, $\bar{\boldsymbol{\sigma}}_j \triangleq \mathbb{E}_{p(\mathbf{x},\mathbf{u})}[\boldsymbol{\sigma}_j]$, and $\bar{\boldsymbol{\sigma}}_j^\dagger \triangleq \mathbb{E}_{p(\mathbf{u})}[\boldsymbol{\sigma}_j^\dagger]$.*

Using Propositions 2 and 3, we can now show that passive variables of the sampled representation follow the prior distribution while active variables are close to their corresponding mean representation, which extends the corresponding proof for standard VAEs (Bonheme & Grzes, 2021).

**Theorem 2** (Polarised regime of $\mathbf{z}$ over $\boldsymbol{X}$ and $\boldsymbol{U}$). *When an iVAE learns in a polarised regime, its sampled representation $\mathbf{z}$ is composed of a set of passive, active and mixed variables $\mathbb{V}_p \cup \mathbb{V}_a \cup \mathbb{V}_m$ such that, over $\boldsymbol{X}$ and $\boldsymbol{U}$:*

*(i)* $p(\mathbf{z}_j) \approx p(\mathbf{z}_j^\dagger)$ $\quad \forall\, j \in \mathbb{V}_p$,

*(ii)* $p(\mathbf{z}_j) \approx p(\boldsymbol{\mu}_j)$ $\quad \forall\, j \in \mathbb{V}_a$,

*(iii)* $p(\mathbf{z}_j) = c\, p(\mathbf{z}_j^\dagger) + (1 - c)\, p(\boldsymbol{\mu}_j)$ $\quad \forall\, j \in \mathbb{V}_m$, *where $0 < c < 1$.*

## 3 Conclusion

We have shown that the polarised regime of standard VAEs can be seen as a specific case of the polarised regime of iVAEs where the mean and variance of the prior are fixed to $\mathbf{0}$ and $\boldsymbol{I}$. Thus, as for standard VAEs, the mean representations of iVAEs should be pruned of their passive variables when used on downstream tasks (Bonheme & Grzes, 2021). Furthermore, iVAEs are likely to be sensitive to posterior collapse when the pressure on the KL divergence is too high (Dai et al., 2020). While these results can be generalised to VAEs with any Gaussian prior and posterior with diagonal covariances, extending this work to other prior and posterior distributions is left for future work.

URM STATEMENT

Author Lisa Bonheme meets the URM criteria of the ICLR 2023 Tiny Papers Track.

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

## A PROOFS

### A.1 PROOF OF THEOREM 1

We want to show that a well-behaved iVAE—an iVAE converging to the lowest reconstruction error and KL divergence possible—learns in a polarised regime. That is, any superfluous (passive) latent dimension is discarded and relevant (active) variables are learned with high precision. Specifically, passive variables will only depend on the prior while active variables will have very low variance. Based on Dai & Wipf (2018, Theorem 5), we will show that iVAEs similarly behave in a polarised regime.

*Proof.* Let us consider the following multivariate Gaussian distributions with diagonal covariance

$$q_{\boldsymbol{\phi}}(\mathbf{z}|\mathbf{x}, \mathbf{u}) = \mathcal{N}(\boldsymbol{\mu}, \boldsymbol{\Sigma}), \tag{1}$$

$$p_{\boldsymbol{\theta}}(\mathbf{z}|\mathbf{u}) = \mathcal{N}(\boldsymbol{\mu}^{\dagger}, \boldsymbol{\Sigma}^{\dagger}), \tag{2}$$

$$p_{\boldsymbol{\theta}}(\mathbf{x}|\mathbf{z}) = \mathcal{N}(\boldsymbol{\mu}^{\ddagger}, \gamma \boldsymbol{I}), \tag{3}$$

where $\boldsymbol{\mu}^{\ddagger} \stackrel{\text{def}}{=} \boldsymbol{\mu}^{\ddagger}[\mathbf{z}; \theta]$ and $\gamma > 0$. Given $k$ latent dimensions, the learning objective to minimise is

$$-\mathcal{L}(\boldsymbol{\theta}, \boldsymbol{\phi}; \mathbf{x}, \mathbf{u}) = -\mathbb{E}_{q_{\boldsymbol{\phi}}(\mathbf{z}|\mathbf{x}, \mathbf{u})}[\log p_{\boldsymbol{\theta}}(\mathbf{x}|\mathbf{z}) + D_{\text{KL}}(q_{\boldsymbol{\phi}}(\mathbf{z}|\mathbf{x}, \mathbf{u}) \parallel p_{\boldsymbol{\theta}}(\mathbf{z}|\mathbf{u})). \tag{4}$$

Plugging Equation 3 into Equation 4, and dropping the dependencies on model parameters for readability we get

$$-\mathcal{L}(\boldsymbol{\theta}, \boldsymbol{\phi}; \mathbf{x}, \mathbf{u}) = \frac{1}{2\gamma} \mathbb{E}_{q_{\boldsymbol{\phi}}(\mathbf{z}|\mathbf{x}, \mathbf{u})} \left[ \|\mathbf{x} - \boldsymbol{\mu}^{\ddagger}[\mathbf{z}]\|_2^2 \right] + \frac{k}{2} \log(2\pi\gamma) + D_{\text{KL}}(q_{\boldsymbol{\phi}}(\mathbf{z}|\mathbf{x}, \mathbf{u}) \parallel p_{\boldsymbol{\theta}}(\mathbf{z}|\mathbf{u})) \tag{5}$$

$$\geqslant \frac{1}{2\gamma} \mathbb{E}_{p(\boldsymbol{\epsilon})} \left[ \|\mathbf{x} - \boldsymbol{\mu}^{\ddagger}[\boldsymbol{\mu} + \boldsymbol{\epsilon}\boldsymbol{\sigma}^{1/2}]\|_2^2 \right] + \frac{k}{2} \log(2\pi\gamma). \tag{6}$$

Now suppose that the reconstruction is highly precise (i.e., $\gamma \to 0$), and we have

$$\lim_{\gamma \to 0} \mathbb{E}_{p(\boldsymbol{\epsilon})} \left[ \|\mathbf{x} - \boldsymbol{\mu}^{\ddagger}[\boldsymbol{\mu} + \boldsymbol{\epsilon}\boldsymbol{\sigma}^{1/2}]\|_2^2 \right] = \Delta.$$

If $\Delta \neq 0$, we have $\lim_{\gamma \to 0} -\mathcal{L}(\boldsymbol{\theta}, \boldsymbol{\phi}; \mathbf{x}) \geqslant \lim_{\gamma \to 0} \frac{\Delta}{2\gamma} + \frac{k}{2} \log(2\pi\gamma) = +\infty$ which contradicts the fact that $-\mathcal{L}(\boldsymbol{\theta}, \boldsymbol{\phi}; \mathbf{x})$ converges to $-\infty$ (i.e., the model is not well-behaved). Thus we must have $\Delta = 0$ and, given that $\|\mathbf{x} - \boldsymbol{\mu}^{\ddagger}[\boldsymbol{\mu} + \boldsymbol{\epsilon}\boldsymbol{\sigma}^{1/2}]\|_2^2 \geqslant 0$, it means that

$$\lim_{\gamma \to 0} \boldsymbol{\mu}^{\ddagger}[\boldsymbol{\mu} + \boldsymbol{\epsilon}\boldsymbol{\sigma}^{1/2}] = \mathbf{x}.$$

Furthermore, if $\boldsymbol{\epsilon} = 0$, this becomes

$$\lim_{\gamma \to 0} \boldsymbol{\mu}^{\ddagger}[\boldsymbol{\mu}] = \mathbf{x}. \tag{7}$$

We will now see that this is achieved by setting $\boldsymbol{\sigma}$ to very low values on active variables.

Recalling Equation 7, let us derive the Taylor approximation of $\boldsymbol{\mu}^{\ddagger}[\mathbf{z}]$ at $\mathbf{z} = \boldsymbol{\mu}$

$$\boldsymbol{\mu}^{\ddagger}[\mathbf{z}] \approx \boldsymbol{\mu}^{\ddagger}[\boldsymbol{\mu}] + \boldsymbol{\mu}^{\ddagger'}[\boldsymbol{\mu}](\mathbf{z} - \boldsymbol{\mu}) \approx \mathbf{x} + \boldsymbol{\mu}^{\ddagger'}[\boldsymbol{\mu}](\mathbf{z} - \boldsymbol{\mu}). \tag{8}$$

Plugging Equation 8 into Equation 5, and letting $C \stackrel{\text{def}}{=} \frac{k}{2} \log(2\pi\gamma) + D_{\text{KL}}(q_{\boldsymbol{\phi}}(\mathbf{z}|\mathbf{x}, \mathbf{u}) \parallel p_{\boldsymbol{\theta}}(\mathbf{z}|\mathbf{u}))$, we get

$$-\mathcal{L}(\boldsymbol{\theta}, \boldsymbol{\phi}; \mathbf{x}, \mathbf{u}) \approx \frac{1}{2\gamma} \mathbb{E}_{q_{\boldsymbol{\phi}}(\mathbf{z}|\mathbf{x}, \mathbf{u})} \left[ \|\mathbf{x} - \mathbf{x} - \boldsymbol{\mu}^{\ddagger'}[\boldsymbol{\mu}](\mathbf{z} - \boldsymbol{\mu})\|_2^2 \right] + C,$$

$$= \frac{1}{2\gamma} \mathbb{E}_{q_{\boldsymbol{\phi}}(\mathbf{z}|\mathbf{x}, \mathbf{u})} \left[ \left( \boldsymbol{\mu}^{\ddagger'}[\boldsymbol{\mu}](\mathbf{z} - \boldsymbol{\mu}) \right)^T \boldsymbol{\mu}^{\ddagger'}[\boldsymbol{\mu}](\mathbf{z} - \boldsymbol{\mu}) \right] + C,$$

$$= \frac{1}{2\gamma} \mathbb{E}_{q_{\boldsymbol{\phi}}(\mathbf{z}|\mathbf{x}, \mathbf{u})} \left[ Tr \left( (\mathbf{z} - \boldsymbol{\mu})^T (\boldsymbol{\mu}^{\ddagger'}[\boldsymbol{\mu}])^T \boldsymbol{\mu}^{\ddagger'}[\boldsymbol{\mu}](\mathbf{z} - \boldsymbol{\mu}) \right) \right] + C,$$

$$= \frac{1}{2\gamma} Tr \left( \mathbb{E}_{q_{\boldsymbol{\phi}}(\mathbf{z}|\mathbf{x}, \mathbf{u})}[(\mathbf{z} - \boldsymbol{\mu})^T (\mathbf{z} - \boldsymbol{\mu})] (\boldsymbol{\mu}^{\ddagger'}[\boldsymbol{\mu}])^T \boldsymbol{\mu}^{\ddagger'}[\boldsymbol{\mu}] \right) + C,$$

$$= \frac{1}{2\gamma} Tr \left( \boldsymbol{\Sigma}(\boldsymbol{\mu}^{\ddagger'}[\boldsymbol{\mu}])^T \boldsymbol{\mu}^{\ddagger'}[\boldsymbol{\mu}] \right) + C. \tag{9}$$

Plugging Equations 1 and 2 into the KL divergence term of Equation 5, we get

$$-\mathcal{L}(\boldsymbol{\theta}, \boldsymbol{\phi}; \mathbf{x}, \mathbf{u}) = \frac{1}{2\gamma} Tr \left( \boldsymbol{\Sigma}(\boldsymbol{\mu}^{\ddagger'}[\boldsymbol{\mu}])^T \boldsymbol{\mu}^{\ddagger'}[\boldsymbol{\mu}] \right) + \frac{1}{2} Tr(\boldsymbol{\Sigma}^{\dagger^{-1}} \boldsymbol{\Sigma}) + \frac{k}{2} \log(2\pi\gamma) +$$

$$\frac{1}{2} \left( (\boldsymbol{\mu}^{\dagger} - \boldsymbol{\mu})^T \boldsymbol{\Sigma}^{\dagger^{-1}} (\boldsymbol{\mu}^{\dagger} - \boldsymbol{\mu}) - k + \log|\boldsymbol{\Sigma}^{\dagger}| - \log|\boldsymbol{\Sigma}| \right),$$

$$= \frac{1}{2} Tr \left( \boldsymbol{\Sigma} \left( \boldsymbol{\Sigma}^{\dagger^{-1}} + \frac{1}{\gamma} (\boldsymbol{\mu}^{\ddagger'}[\boldsymbol{\mu}])^T \boldsymbol{\mu}^{\ddagger'}[\boldsymbol{\mu}] \right) \right) + \frac{k}{2} \log(2\pi\gamma) +$$

$$\frac{1}{2} \left( (\boldsymbol{\mu}^{\dagger} - \boldsymbol{\mu})^T \boldsymbol{\Sigma}^{\dagger^{-1}} (\boldsymbol{\mu}^{\dagger} - \boldsymbol{\mu}) - k + \log|\boldsymbol{\Sigma}^{\dagger}| - \log|\boldsymbol{\Sigma}| \right). \tag{10}$$

The optimal value of $\boldsymbol{\Sigma}$ must thus satisfy

$$\boldsymbol{\Sigma} = \left(\boldsymbol{\Sigma}^{\dagger^{-1}} + \frac{1}{\gamma}(\boldsymbol{\mu}^{\ddagger'}[\boldsymbol{\mu}])^T\boldsymbol{\mu}^{\ddagger'}[\boldsymbol{\mu}]\right)^{-1}. \tag{11}$$

As $\boldsymbol{\mu}^{\ddagger'}[\boldsymbol{\mu}]$ is a tangent space of the $r$-dimensional manifold $\mathcal{X}$ at $\boldsymbol{\mu}^{\ddagger}[\boldsymbol{\mu}]$, it has a rank of $r$. It follows that $(\boldsymbol{\mu}^{\ddagger'}[\boldsymbol{\mu}])^T\boldsymbol{\mu}^{\ddagger'}[\boldsymbol{\mu}]$ can be decomposed as $\boldsymbol{U}^{\ddagger}\boldsymbol{S}^{\ddagger}\boldsymbol{S}^{\ddagger^T}\boldsymbol{U}^{\ddagger^T}$ where the first $r$ elements of the diagonal matrix $\boldsymbol{S}^{\ddagger}\boldsymbol{S}^{\ddagger^T} \in \mathbb{R}^{k\times k}$ are nonzero, such that $diag[\boldsymbol{S}^{\ddagger}\boldsymbol{S}^{\ddagger^T}] = [\lambda_1^{\ddagger}, \lambda_2^{\ddagger}, \cdots, \lambda_r^{\ddagger}, 0, \cdots, 0]$. If $r = k$, given that $\boldsymbol{\Sigma}$ and $(\boldsymbol{\mu}^{\ddagger'}[\boldsymbol{\mu}])^T\boldsymbol{\mu}^{\ddagger'}[\boldsymbol{\mu}]$ are $k$-by-$k$ symmetric matrices, by Golub & Van Loan (2013, Theorem 8.1.5), we have

$$\frac{1}{\frac{1}{\lambda_{min}^{\dagger}} + \frac{\lambda_i^{\ddagger}}{\gamma}} \leqslant \lambda_i \leqslant \frac{1}{\frac{1}{\lambda_{max}^{\dagger}} + \frac{\lambda_i^{\ddagger}}{\gamma}} \qquad \forall i = 1, \cdots, k. \tag{12}$$

We can directly see that as $\gamma \to 0$, both sides of the inequality converge to 0, thus the eigenvalues of $\boldsymbol{\Sigma}$ become arbitrarily small at a rate proportional to $\gamma$. Thus similarly to Dai & Wipf (2018), $\frac{1}{\sqrt{\gamma}}\boldsymbol{\Sigma}^{1/2} \approx O(1)$ under mild conditions and around the optimal solution, we have

$$-2\mathbb{E}_{q_\phi(\mathbf{z}|\mathbf{x},\mathbf{u})}[\log p_\theta(\mathbf{x}|\mathbf{z})] = 2\mathbb{E}_{q_\phi(\mathbf{z}|\mathbf{x},\mathbf{u})}\left[\frac{1}{\gamma}\|\mathbf{x} - \boldsymbol{\mu}^{\ddagger}[\mathbf{z}]\|_2^2\right] + k\log(2\pi\gamma), \tag{13}$$

$$\approx \mathbb{E}_{q_\phi(\mathbf{z}|\mathbf{x},\mathbf{u})}[O(1)] + k\log(2\pi\gamma), \tag{14}$$

$$= O(1) + k\log\gamma. \tag{15}$$

Moreover, because $2\mathbb{E}_{q_\phi(\mathbf{z}|\mathbf{x},\mathbf{u})}[\log p_\theta(\mathbf{x}|\mathbf{z})] \geqslant 0$, one can see that the lower bound provided by Equation 15 cannot be further decreased. Thus, any additional variables introduced by increasing $k$ will not improve the reconstruction but may have a negative impact on the KL divergence when $\gamma \to 0$. Thus, similarly to VAEs, any superfluous variable of iVAEs will seek to lower the cost of the KL divergence by remaining close to the prior. $\qquad\square$

## A.2 PROOF OF PROPOSITION 1

*Proof.* Let us consider the classical case where the prior is a standard Gaussian distribution. Following Rolinek et al. (2019); Bonheme & Grzes (2021), we have

**Definition 1** (Polarised regime of VAEs). *When a VAE learns in a polarised regime, its mean, variance, and sampled representations, $\boldsymbol{\mu}^{(i)}$, $\boldsymbol{\sigma}^{(i)}$, and $\mathbf{z}^{(i)}$, are composed of a set of passive and active variables, $\mathbb{V}_p^{(i)} \cup \mathbb{V}_a^{(i)}$ such that, for each data example $\mathbf{x}^{(i)}$:*

*(i)* $\boldsymbol{\mu}_j^{(i)} \approx 0$, $\boldsymbol{\sigma}_j^{(i)} \approx 1$*, and* $\mathbf{z}_j^{(i)} \approx \boldsymbol{\epsilon}_j^{(i)} \quad \forall j \in \mathbb{V}_p^{(i)}$,

*(ii)* $\boldsymbol{\sigma}_j^{(i)} \ll 1$ *and* $\mathbf{z}_j^{(i)} \approx \boldsymbol{\mu}_j^{(i)} \quad \forall j \in \mathbb{V}_a^{(i)}$,

*where $\boldsymbol{\epsilon}^{(i)} \sim \mathcal{N}(\mathbf{0}, \boldsymbol{I})$, and $j$ is the $j^{th}$ variable of a representation.*

Statement (i) comes from the fact that the prior is $\mathcal{N}(\mathbf{0}, \boldsymbol{I})$ and passive variables are as close as possible of the prior to decrease the KL divergence, while statement (ii) shows that active variables have high precision (i.e., low variance) while increasing the KL divergence. For iVAEs, statement (ii) remains unchanged but statement (i) needs to be updated to take into account the new prior distribution $\mathcal{N}(\boldsymbol{\mu}^{\dagger}, diag[\boldsymbol{\sigma}^{\dagger}])$.

Because the prior distribution is $\mathcal{N}(\boldsymbol{\mu}^{\dagger}, diag[\boldsymbol{\sigma}^{\dagger}])$, $\mathbf{z}^{\dagger} = \boldsymbol{\mu}^{\dagger} + \boldsymbol{\epsilon}\left(\boldsymbol{\sigma}^{\dagger}\right)^{1/2}$ where $\boldsymbol{\epsilon} \sim \mathcal{N}(\mathbf{0}, \boldsymbol{I})$. Thus, for passive variables to be as close as possible to the prior, we need $\boldsymbol{\mu}_j^{(i)} \approx \boldsymbol{\mu}_j^{\dagger(i)}$ and $\boldsymbol{\sigma}_j^{(i)} \approx \boldsymbol{\sigma}_j^{\dagger(i)}$ for all $j \in \mathbb{V}_p^{(i)}$. We can thus generalise statement (i) to $\boldsymbol{\mu}_j^{(i)} \approx \boldsymbol{\mu}_j^{\dagger}$, $\boldsymbol{\sigma}_j^{(i)} \approx \boldsymbol{\sigma}_j^{\dagger}$, and $\mathbf{z}_j^{(i)} \approx \mathbf{z}_j^{\dagger(i)} \quad \forall j \in \mathbb{V}_p^{(i)}$, as required. $\qquad\square$

## A.3 Proof of Proposition 2

*Proof.* Let us first consider statement (i) of Proposition 2, where the mean of the prior is learned. We know from Proposition 1 that for all $j \in \mathbb{V}_p^{(i)}$, $\boldsymbol{\mu}_j^{(i)} \approx \boldsymbol{\mu}_j^{\dagger(i)}$. Thus, $\frac{1}{n} \sum_{i=1}^n \boldsymbol{\mu}_j^{(i)} \approx \frac{1}{n} \sum_{i=1}^n \boldsymbol{\mu}_j^{\dagger(i)}$ and $\bar{\boldsymbol{\mu}}_j \approx \bar{\boldsymbol{\mu}}_j^\dagger$. Similarly $\frac{1}{n} \sum_{i=1}^n (\boldsymbol{\mu}_j^{(i)} - \bar{\boldsymbol{\mu}}_j^{(i)})^2 \approx \frac{1}{n} \sum_{i=1}^n (\boldsymbol{\mu}_j^{\dagger(i)} - \bar{\boldsymbol{\mu}}_j^{\dagger(i)})^2$ and $Var(\boldsymbol{\mu}_j) \approx Var(\boldsymbol{\mu}_j^\dagger)$, as required.

In statement (ii), the prior has a fixed mean $\mu^\dagger$. Thus, $\frac{1}{n} \sum_{i=1}^n \boldsymbol{\mu}_j^{(i)} \approx \frac{1}{n} \sum_{i=1}^n \mu_j^\dagger = \mu_j^\dagger$ and $\bar{\boldsymbol{\mu}}_j \approx \mu_j^\dagger$. Similarly $\frac{1}{n} \sum_{i=1}^n (\boldsymbol{\mu}_j^{(i)} - \bar{\boldsymbol{\mu}}_j^{(i)})^2 \approx \frac{1}{n} \sum_{i=1}^n (\mu_j^\dagger - \mu_j^\dagger)^2 \ll 1$ and $Var(\boldsymbol{\mu}_j) \ll 1$, as required. $\qquad\square$

## A.4 Proof of Proposition 3

*Proof.* Let us first consider statement (i) of Proposition 3 which concerned the passive variables of the variance representation. We know from Proposition 1 that for all $j \in \mathbb{V}_p^{(i)}$, $\boldsymbol{\sigma}_j^{(i)} \approx \boldsymbol{\sigma}_j^{\dagger(i)}$. Thus, $\frac{1}{n} \sum_{i=1}^n \boldsymbol{\sigma}_j^{(i)} \approx \frac{1}{n} \sum_{i=1}^n \boldsymbol{\sigma}_j^{\dagger(i)}$ and $\bar{\boldsymbol{\sigma}}_j \approx \bar{\boldsymbol{\sigma}}_j^\dagger$. Similarly $\frac{1}{n} \sum_{i=1}^n (\boldsymbol{\sigma}_j^{(i)} - \bar{\boldsymbol{\sigma}}_j^{(i)})^2 \approx \frac{1}{n} \sum_{i=1}^n (\boldsymbol{\sigma}_j^{\dagger(i)} - \bar{\boldsymbol{\sigma}}_j^{\dagger(i)})^2$ and $Var(\boldsymbol{\sigma}_j) \approx Var(\boldsymbol{\sigma}_j^\dagger)$, as required.

Statement (ii) of Proposition 3 is concerned with the active variables of the variance representation. We know from Proposition 1 that for all $j \in \mathbb{V}_a^{(i)}$, $\boldsymbol{\sigma}_j^{(i)} \ll 1$. Thus, $\frac{1}{n} \sum_{i=1}^n \boldsymbol{\sigma}_j^{(i)} \ll 1$ and $\bar{\boldsymbol{\sigma}}_j \ll 1$. Similarly $\frac{1}{n} \sum_{i=1}^n (\boldsymbol{\sigma}_j^{(i)} - \bar{\boldsymbol{\sigma}}_j^{(i)})^2 \ll 1$ and $Var(\boldsymbol{\sigma}_j) \ll 1$, as required.

$\qquad\square$

## A.5 Proof of Theorem 2

*Proof.* Let $\mathbf{z}_j$ be the sampled representation variable at index $j$. There are three cases:

(i) If $j \in \mathbb{V}_p$, then, from statement (i) of Proposition 2, $\boldsymbol{\mu}_j \approx \boldsymbol{\mu}_j^\dagger$. Moreover, from statement (i) of Proposition 2, $\boldsymbol{\sigma}_j \approx \boldsymbol{\sigma}_j^\dagger$. Thus, $\mathbf{z}_j \approx \boldsymbol{\mu}_j^\dagger + \boldsymbol{\epsilon}_j \left(\boldsymbol{\sigma}_j^\dagger\right)^{1/2}$. Recall from Proposition 1 that $\mathbf{z}_j^\dagger$ is distributed according to $N(\boldsymbol{\mu}^\dagger, diag[\boldsymbol{\sigma}^\dagger])$. It follows that $p(\mathbf{z}_j) \approx p(\mathbf{z}_j^\dagger)$, which proves statement (i).

(ii) If $j \in \mathbb{V}_a$, then, from statement (ii) of Proposition 3, $\boldsymbol{\sigma}_j$ is almost constant with a value close to 0. Thus, $\mathbf{z}_j \approx \boldsymbol{\mu}_j$. It follows that $p(\mathbf{z}_j) \approx p(\boldsymbol{\mu}_j)$, which proves statement (ii).

(iii) If $j \in \mathbb{V}_m$, we know that $\mathbf{z}_j$ is composed of a subset of active components and a subset of passive components. Thus, $\mathbf{z}_j$ is distributed according to a mixture distribution. Using step (i) and (ii) of the proof, we know that $p(\mathbf{z}_j) \approx p(\mathbf{z}_j^\dagger)$ for passive variables and $p(\mathbf{z}_j) \approx p(\boldsymbol{\mu}_j)$ for active variables. It follows that for mixed variables $p(\mathbf{z}_j) = c \; p(\mathbf{z}_j^\dagger) + (1 - c) \; p(\boldsymbol{\mu}_j)$ where $0 < c < 1$. This concludes the proof.

$\qquad\square$

## B Empirical verification

In this section, we provide an empirical verification of the propositions and theorems presented in the main paper. The source code is available at `https://github.com/bonheml/VAE_learning_dynamics`, the architecture and hyperparameters used are described in Table 1 and 2. We use the dSprites dataset[1] (Higgins et al., 2017). Note that all the histograms presented below are computed using 10000 input examples and the latent space is set to a larger number of dimensions than usual (30 instead of 10) to ensure the presence of superfluous (passive) variables. In line with the original implementation of Khemakhem et al. (2020), we do not learn the mean representation

---

[1] Licensed under an Apache 2.0 licence.

and fix it to 0. Note that as with VAEs, we did not observe any mixed variables when training iVAEs on this dataset.

Table 1: Model architecture

| Encoder |
| --- |
| Input: $(\mathbb{R}^{64 \times 64 \times channels}, \mathbb{R}^5)$ |
| Conv, kernel=4×4, filters=32, activation=ReLU, strides=2 |
| Conv, kernel=4×4, filters=32, activation=ReLU, strides=2 |
| Conv, kernel=4×4, filters=64, activation=ReLU, strides=2 |
| Conv, kernel=4×4, filters=64, activation=ReLU, strides=2 |
| FC, output shape=261, activation=ReLU |
| FC, output shape=2x30 |

| Decoder |
| --- |
| Input: $\mathbb{R}^{30}$ |
| FC, output shape=256, activation=ReLU |
| Deconv, kernel=4×4, filters=64, activation=ReLU, strides=2 |
| Deconv, kernel=4×4, filters=32, activation=ReLU, strides=2 |
| Deconv, kernel=4×4, filters=32, activation=ReLU, strides=2 |
| Deconv, kernel=4×4, filters=channels, activation=ReLU, strides=2 |

| Prior variance |
| --- |
| Input: $\mathbb{R}^5$ |
| FC, output shape=50, activation=Leaky ReLU |
| FC, output shape=50, activation=Leaky ReLU |
| FC, output shape=50, activation=Leaky ReLU |
| FC, output shape=30 |

Table 2: Model hyperparameters

| Parameter | Value |
| --- | --- |
| Batch size | 64 |
| Latent space dimension | 30 |
| Optimizer | Adam |
| Adam: $\beta_1$ | 0.9 |
| Adam: $\beta_2$ | 0.999 |
| Adam: $\epsilon$ | 1e-8 |
| Adam: learning rate | 0.0001 |
| Reconstruction loss | Bernoulli |
| Training steps | 300,000 |
| Train/test split | 90/10 |

**Illustration of Proposition 2**   We can see in Figure 1a that the empirical distribution of a passive variable of the mean representation consistently takes values very close to zero, which confirms Proposition 2 for the configuration where the mean is fixed. Moreover, the active variables of the mean representation tend to have higher variance as they encode more information, as illustrated in Figure 2a.

**Illustration of Proposition 3**   Figure 2b shows that the active variables of the variance representation will remain close to zero. Moreover, when compared to Figure 2c, we can see that the active variables of the variance representation also depart from their prior distribution, as their objective is to maximise the reconstruction by reducing the noise during the reparametrisation. On the other hand, the passive variables of the variance representation stay close to the variance representation of the prior to minimise the KL divergence, as seen in Figures 1b and 1c. Both observations confirm Proposition 3. Interestingly, in the case of passive variables, the variance representation of the

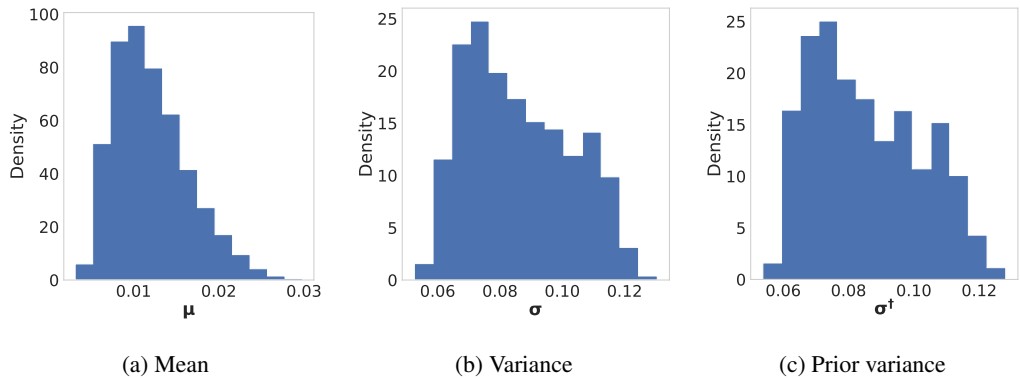

Figure 1: Empirical distributions of a passive variable of an iVAE trained on dSprites. (a) and (b) correspond to the mean and variance of the posterior, and (c) is the variance of the prior. The same passive variable is used for all plots in this figure.

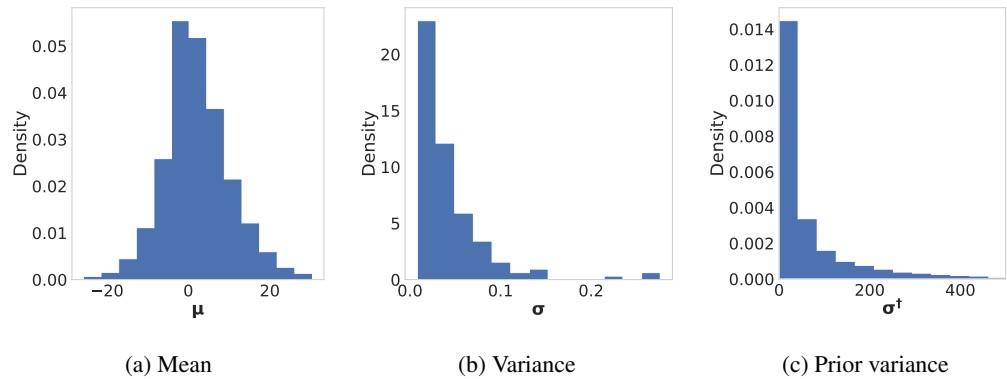

Figure 2: Empirical distributions of an active variable of an iVAE trained on dSprites. (a) and (b) correspond to the mean and variance of the posterior, and (c) is the variance of the prior. The same active variable is used for all plots in this figure.

prior stays close to zero. Thus, iVAEs seems to behave in a near-deterministic way as the variance is maintained low for both types of variables.

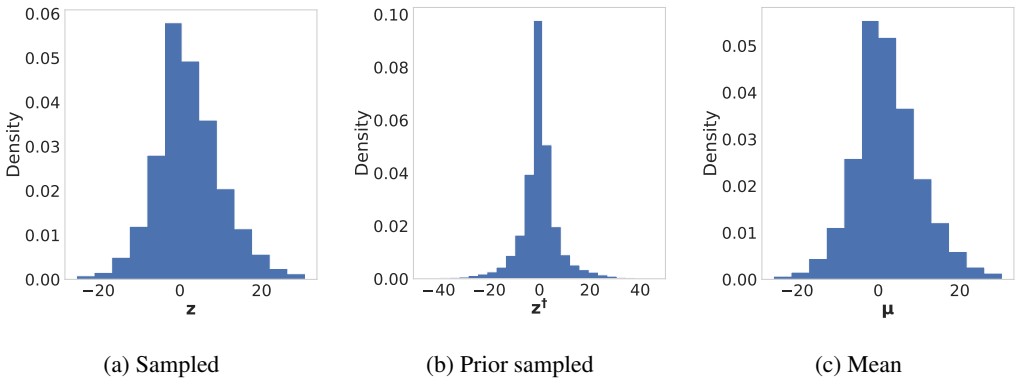

Figure 3: Empirical distributions of an active variable of an iVAE trained on dSprites. (a) and (b) correspond to the sampled representation of the posterior and the prior. (c) is the mean representation of the posterior. The same active variable is used for all plots in this figure.

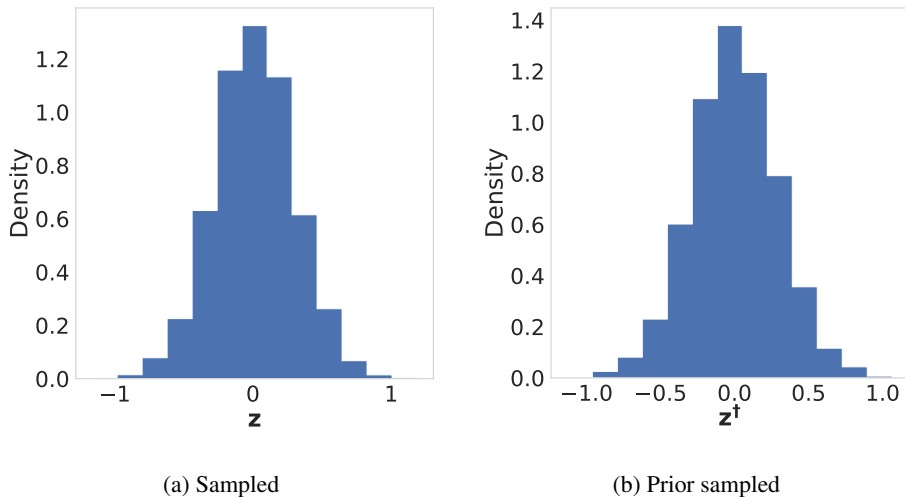

(a) Sampled  (b) Prior sampled

Figure 4: Empirical distributions of a passive variable of an iVAE trained on dSprites. (a) and (b) correspond to the sampled representation of the posterior and the prior. The same passive variable is used for all plots in this figure.

**Illustration of Theorem 2**  Figure 3 confirms that the sampled representations of active variables are very close to the corresponding mean representation, as illustrated by the very similar empirical distributions of Figures 3a and 3c. We can also observe a strong difference between the sampled representation of the posterior and the prior in Figures 3a and 3b. The second point of Theorem 2 is confirmed by Figure 4 where we can see that the sampled representations of the prior and posterior are very similar.

