# OpenReview forum: "The Polarised Regime of identifiable Variational Autoencoders"
_ICLR.cc/2023/TinyPapers — Submitted to Tiny Papers @ ICLR 2023_

### Official Review · Reviewer_nQSQ · 2023-03-24

**Confidence:** 4

**Summary Of Contributions:**

Descriptions of math theorems -- I encourage you to improve on clarity if you have a longer version

**Rating:**

Great Start (GS): a submission which meets some of the reviewing criteria but has room for improvement

**Strengths And Weaknesses:**

This paper provides theoretical results to show whether iVAE in training also has behaviours that can be classified into the polarised regime.

Just to say that in quite some other papers this phenomenon is called "posterior collapse", see:
https://arxiv.org/abs/1901.03416

I believe the results in the paper are correct (as they are extensions to conditional vat with Gaussian prior conditioned on u).

The main issue for this paper is that the presentation is a bit too dry -- it's a collection of math theorem text with very little explanation. The paper starts from background that listed existing work with little discussions. See my comments below for my suggested changes.

**Suggested Changes:**

I encourage you to consider the following changes:
1. Motivate what is "polarised regime" and why people are interested in studying it;
2. Why studying this problem for iVAE is interesting: how would it impact on the specific claims (e.g., practical identifiability) of iVAE?
3. Does your result extend to general exponential family for the conditional prior?
4. Translate your theorem results (in math notations) to "layman text", e.g., if Theorem 2 holds, then what's the indication in practical iVAE training?

---

> ### Author Response · Authors · 2023-04-22
> **Answer to reviewer nQSQ**
>
> Thank you for your interest in our work and for your detailed comments.
>
> Answers to the reviewer's questions:
> -------------------------------------------------
> > Just to say that in quite some other papers this phenomenon is called "posterior collapse", see: https://arxiv.org/abs/1901.03416
>
> While mathematically related, the polarised regime and posterior collapse are two different concepts, posterior collapse being a pathologic case of the desirable polarised regime.
> Let us consider a VAE learning in a polarised regime whose latent representations are composed of 5 active variables and 5 passive variables.
> The decoder will only use the 5 active variables and still generate reconstructed images of good quality.
> If the same model was a victim of posterior collapse, it would instead be composed of 10 passive variables, and because the decoder would not be able to access any information from the input, the reconstruction quality would be poor.
> For this reason the polarised regime is also sometimes called selective posterior collapse (e.g., in [1]).
>
> > Motivate what is "polarised regime" and why people are interested in studying it
>
> We have now added a paragraph before the background section to explain the motivation of this paper. Unfortunately, we cannot do a fully detailed introduction due to
> space constraints.
>
> > Why studying this problem for iVAE is interesting: how would it impact on the specific claims (e.g., practical identifiability) of iVAE?
>
> We have added some details about this in the conclusion. Basically, the polarised regime results in a very typical behaviour in VAEs.
> For example, the passive variables of the mean representation tend to be correlated with the active variables [2]. This can pose stability and interpretability issues
> when those representations are used as input to models that are sensitive to multicollinearity (e.g., regression models).
> Because iVAEs are identifiable and thus provide transparent embeddings, they are very attractive to use on downstream tasks.
> However, because they also learn in a polarised regime, they may suffer from the same correlation issues.
> Knowing our results, the practitioners should be encouraged to remove passive variables from the mean representations before using them on downstream tasks to improve downstream task's performance.
>
>
> > Does your result extend to general exponential family for the conditional prior?
>
> We have added a discussion about this in the conclusion.
> Our result generalises to any VAE with a Gaussian prior and posterior.
> However, other distributions would not fit the assumptions of Sec.2, so we cannot readily extend our results to such types of models.
> This is because the proof of Theorem 1 relies on the closed form KL divergence between Gaussians and the reparametrisation
> trick, which may both change significantly for other distributions (even from the exponential family).
>
> > Translate your theorem results (in math notations) to "layman text", e.g., if Theorem 2 holds, then what's the indication in practical iVAE training?
>
> We now have an explanatory sentence before each properties and theorems to describe what each result entails.
>
> References
> ----------
> [1] Dai, B., Wang, Z., & Wipf, D. (2020, November). The usual suspects? Reassessing blame for VAE posterior collapse.
> In International conference on machine learning (pp. 2313-2322). PMLR.
>
> [2] Bonheme, L., & Grzes, M. (2021). Be More Active! Understanding the Differences between Mean and Sampled
> Representations of Variational Autoencoders. arXiv preprint arXiv:2109.12679.

---

### Official Review · Reviewer_k7Cc · 2023-03-30

**Confidence:** 3

**Summary Of Contributions:**

The polarized regime, i.e., is well-studied in the context of vanilla VAEs. In this paper, the authors have shown that iVAE, where the mean and the variance of the prior are learned, behave in a polarized regime, and display similar active variables as standard VAEs, whose priors are fixed. Finally, they have shown that the properties of “passive variables” of standard VAEs are a specific case of those of iVAE where the mean and variance of the prior are fixed to 0 and I.

**Rating:**

High Potential (HP): a submission which meets the reviewing criteria and has potential to make an impact on the field

**Strengths And Weaknesses:**

The contribution of the paper is clearly presented, but needs some fixes regarding English and grammatical aspects. In the Appendix are all the necessary demonstrations relative to the theorems and prepositions stated. The article, therefore, turns out to be correct and reproducible from a theoretical point of view.

**Suggested Changes:**

It would have been nice to see one experimental result.
Finally, the paper could provide information about the limitation of the proposed approach.

---

> ### Author Response · Authors · 2023-04-22
> **Answer to reviewer k7Cc**
>
> Thank you for your interest in our work and for your detailed comments.
>
> Answers to the reviewer's questions:
> -------------------------------------------------
>
> > It would have been nice to see one experimental result
>
> We have now added an empirical verification of the propositions and theorems of the main paper in Appendix B.
>
> > The paper could provide information about the limitation of the proposed approach.
>
> This is now discussed in the conclusion.

---

### Author Response · Authors · 2023-04-22
**Summary of the revised version**

Dear reviewers,
Thank you for your time and effort.
We have implemented your suggestions and we have uploaded a revised version of our paper to openreview.

Summary of the changes
----------------------------------
- There is now a paragraph explaining the motivation of this paper before the background section.
- The theorems and propositions are summarised to ease the reading.
- The conclusion now provides more details about the impact and limitations of our work.
- We have added a new appendix containing empirical validation of the properties and theorems. Details of the implementation and a link to the code is also shared for reproductibility.
- The paper is deanonymised.

Other comments
----------------------
We would like to point out that while related, the polarised regime and posterior collapse are two different concepts.
The polarised regime is a desirable property of VAEs which ensures that a minimal number of variables is used to encode information, the remaining (passive) dimensions are then collapsed to the prior.
Posterior collapse is a pathologic case of the desirable polarised regime where all the latent variables are passive (e.g., because the KL divergence is too high).
Because the latent representations are uninformative in that case, they would lead to poor reconstruction quality and bad performance when used as embeddings on downstream tasks.

---

### Meta-Review · Area_Chair_6kbW · 2023-04-04

**Recommendation:** Invite to present
**Confidence:** 4

**Metareview:**

Good paper with all reviewers arguing for acceptance. Some changes are suggested.

**Summary:**

The manuscript studies the polarized regime (posterior collapse) in iVAE. The theoretical results are interesting and appear to be solid. Further improvements on the intuitions behind the theorems could be helpful.

**Reason For Not Giving A Higher Recommendation:**

Perhaps more discussion on the significance, intuition, and motivation of the proposed theoretical results could be helpful. In particular, since iVAE majorly focuses on the identifiability of the hidden generating process, some discussion on why the proposed results could benefit the identification in practice could be interesting.

**Reason For Not Giving A Lower Recommendation:**

N/A

---

### Decision · Program_Chairs · 2023-04-10

Invite to present